# Differences in Loneliness and Social Isolation among Community-Dwelling Older Adults by Household Type: A Nationwide Survey in Japan

**DOI:** 10.3390/healthcare11111647

**Published:** 2023-06-04

**Authors:** Nanami Oe, Etsuko Tadaka

**Affiliations:** Department of Community and Public Health Nursing, Faculty of Health Sciences, Hokkaido University, K12-N5, Kita-ku, Sapporo 060-0812, Japan; o_nanami0706@pop.med.hokudai.ac.jp

**Keywords:** community, household type, healthy longevity, loneliness, social isolation

## Abstract

(1) Background: Social isolation and loneliness are determinants of healthy longevity. However, previous research has focused on either social isolation or loneliness and has not considered household types. This study sought to clarify loneliness and social isolation among older adults using single-person (ST) or multi-person (MT) household types. (2) Methods: We administered a national, anonymous, self-administered survey to 5351 Japanese older adults aged 65 years or older. The survey included subjects’ demographic characteristics and scores for loneliness (University of California Los Angeles (UCLA) Loneliness Scale version 3 (Cronbach’s α = 0.790)), social isolation (Lubben Social Network Scale (LSNS-6) (Cronbach’s α = 0.82)), and self-efficacy (GSES). (3) Results: After adjusting for age and gender, ST individuals had significantly lower LSNS-6 and significantly higher UCLA scores than MT individuals (*p* < 0.001). Lower LSNS-6 and higher UCLA scores were significantly associated with lower GSES scores, and the effect of GSES was greater for ST than for MT (LSNS-6, ST (β = 0.358, *p* < 0.001); MT (β = 0.295, *p* < 0.001)) (UCLA, ST (β = −0.476, *p* < 0.001); MT (β = −0.381, *p* < 0.001)). (4) Conclusions: Specific healthcare systems and programs based on self-efficacy should be developed by household type to reduce both social isolation and loneliness.

## 1. Introduction

According to a World Health Organization report, in 2019, the average life expectancy in Japan was longer than that in any other country: 81.5 years for men and 86.9 years for women. However, the average healthy life expectancy in Japan was shorter than the average life expectancy, at 72.6 years for men and 75.5 years for women [1]. Healthy life expectancy is the average period of time during which people have no restrictions in their daily lives [2]. In advanced countries, extending healthy life expectancy is considered to be an important issue as we approach the era of 100-year lifespans. In the Decade of Healthy Aging: Baseline Report, the World Health Organization advocates for the advancement of “the process of developing and maintaining the functional ability that enables well-being in older age,” emphasizing the importance of addressing loneliness and social isolation [3].

There are many theories and models about loneliness and social isolation [4]. One theory is the discrepancy model of loneliness proposed by Perlman and Peplau [5]. They defined predisposing factors for loneliness as individual characteristics and cultural values and norms that—together with precipitating events—may elicit a discrepancy or mismatch between needed, desired, and actual social relations. Depending on individual views and beliefs, people may then experience loneliness and develop different coping reactions. The second theory is the concept of health effects of shared social identity proposed by Haslam et al. Shared social identity may provide independent benefits beyond this, such as buffering against negative social stigma, promoting meaning and connection, allowing positive social influence, and nurturing a sense of collective efficacy [6,7]. Shared social identity has also been linked to reduced loneliness, depression, anxiety, and increased life satisfaction [8,9]. A final theory is the effects of loneliness on health and loneliness reduction approaches proposed by Hawkley et al. [10]. Hawkley et al. proposed an approach for intervening to reduce loneliness. Hawkley’s intervention model emphasizes the importance of influences from society (e.g., awareness campaigns) and social networks/communities (e.g., intergenerational or volunteer programs) on loneliness and their interconnection at the individual level. Of these theories, the theory of Peplau et al. is considered the ‘gold standard’ internationally. It defines loneliness as “the unpleasant experience that occurs when a person’s network of social relationships is deficient in some important way, either quantitatively or qualitatively” [11]. A systematic review of interventions to reduce loneliness in older adults found that the UCLA Loneliness Scale [12], which is based on the theory of Peplau et al., was supported by most articles.

One theory of social isolation is the social network assessment of Lubben [13]. Social networks are used to describe structural aspects of social ties and objective characteristics such as size, frequency, and density [14]. One instrument that is used widely to screen for social isolation in older adults by assessing perceived social support from family and friends is the Lubben Social Network Scale (LSNS) [13,15]. The second theory is the activation of the reaffiliation motive proposed by Qualter et al. [16]. Qualter et al. conceptualized perceived social isolation as a determinant for reaffiliation and motivation to reconnect with other people, which sets the stage for other actions across the lifespan. Perceived social isolation was theorized to be the result of negative affect, which is predetermined, e.g., by views and actions such as behavioral confirmation, social withdrawal, over-attentiveness, and hypervigilance. Another theory is the social safety theory proposed by Slavich [17]. This explains that the primary evolution-based aim of humans is to keep their own bodies biologically and physically safe. Friendly social bonds helped to achieve that aim, and therefore a fundamental human need evolved to create and maintain friendly social bonds. Situations involving social conflict, isolation, devaluation, rejection, and exclusion historically increased the risk of physical injury and infection. An anticipatory neural–immune reactivity to social threat was therefore probably highly conserved. Of these theories, the one proposed by Lubben et al. is considered the ‘gold standard’ internationally. This defines social isolation as “the lack of redundancy of social ties” [15]. A systematic review found that the Lubben Social Network Scale [12], which is based on the theory of Lubben et al., was supported by most articles.

Loneliness and social isolation are different concepts, but they are interrelated research agenda that should be addressed simultaneously [18]. Numerous studies have documented the impact of loneliness and social isolation on the health of older adults. Both loneliness and small social network size have been shown to be associated with mortality risk in older adults [19]. Loneliness and social isolation have also been found to be risk factors for depressive symptomatology [20,21], dementia [22,23], and cardiovascular disease [24,25,26]. As described above, it is clear that loneliness and social isolation in older adults have not only negative impacts on physical and mental health but also the quality of life and healthy longevity. However, few studies have focused on both social isolation and loneliness simultaneously in older adults. It is necessary to address these two simultaneously because they are connected and have a strong impact on the quality of life and healthy longevity of older adults.

According to a basic survey conducted in Japan in 2021, the percentages of respondents in their 60s with scores of 10–12 points (always lonely) and 7–9 points (sometimes lonely) were 4.8% and 34.5%, respectively, in terms of loneliness by age group (3-item Japanese version of the University of California Los Angeles Loneliness Scale version 3). Among those in their 70s, 2.5% of respondents scored 10–12 points (always) and 30.1% scored 7–9 points (sometimes). Among those aged 80 and older, 4.5% scored 10–12 points (always) and 32.2% scored 7–9 points (sometimes). These results indicate that approximately 30% of Japanese older adults suffer from loneliness [27]. A meta-analysis in 2009 revealed that 20%–30% of Europeans aged 65–79 and 40%–50% of those aged 80 and older suffer from loneliness [28]. A study on Japanese older adults reported that 31.5% were socially isolated [29]. In addition, it has been reported that 27.02% of older adults in the United Kingdom [30] and 24% of older adults in the United States are socially isolated [31]. The above findings indicate that loneliness and social isolation are prevalent not only among Japanese older adults but also among older adults in other advanced countries. Therefore, addressing the key public health issues of loneliness and social isolation simultaneously to extend healthy life expectancy is important internationally.

Previous studies have reported demographic factors such as older age [32,33], male gender [31,33,34], low income [31,35,36], being unmarried [31,35], and having a low level of education [31] as factors associated with loneliness and social isolation among older adults. Physical factors such as low subjective health status [35,37] and the presence of chronic diseases [38] have also been reported. Furthermore, social factors, such as a lack of neighborly relationships [35], and environmental factors, such as limited opportunities to participate in social activities [39], have been found to be associated with loneliness and social isolation. As described above, various studies have clarified the factors associated with loneliness and social isolation, but few studies have examined the relationship between loneliness and social isolation and cognitive factors. One variable that can be used as a measure of cognition is self-efficacy. Self-efficacy is an individual’s expectation regarding how well they can perform a necessary action to produce a certain result; thus, self-efficacy is the perception of one’s own potential for action [40]. A study on Chinese adults aged 18 years and older reported that low levels of loneliness were associated with high self-efficacy [41]. A study on Polish adults aged 18 years and older found that self-efficacy and loneliness were significantly correlated [42]. Self-efficacy was strongly associated with social and emotional loneliness in a study on Dutch adults aged 55 years and older (emotional loneliness (β = −0.64, *p* < 0.001); social loneliness (β = −0.56, *p* < 0.001)) [43]. Another study on older adults aged 65 years and over in Taiwan showed that lower self-efficacy tended to be associated with increased loneliness (B = −0.638, *p* < 0.05) [44]. A study on older adults aged 60 years and older living in China found a significant positive correlation between self-efficacy and social networks [45]. Self-efficacy, therefore, has a positive influence on loneliness and social isolation. It is also a modifiable variable, and focusing on it may enable the development of interventions and support to reduce loneliness and social isolation among older adults living in the community. However, there are few good studies on older adults aged 65 years or more.

The strongest predictor for isolation and loneliness in old age is living alone [46,47,48]. According to a survey conducted in Japan, living alone was the most frequently reported event experienced by those with a loneliness score of 7–12 prior to reaching their current loneliness, affecting 20.8% of these individuals [27]. Living alone is an important predictor for social isolation and loneliness. In addition, a study examining factors influencing loneliness among older adults discussed the need to evaluate older adults living alone as a high-risk group [38]. As discussed above, the influence of household type on loneliness and social isolation among older adults is likely to be significant. A study on older widows living alone found that marital happiness acts as a protective factor against loneliness and being the primary caregiver for a spouse prior to widowhood acts as a risk factor for loneliness (loneliness was assessed using the University of California at Los Angeles Loneliness Scale (ULS-8)) [49]. A study on older adults in rural Thailand found that significant differences in age and activities of daily living (ADLs) were associated with social isolation among older people living alone (social isolation was assessed using attended social and physical activity (passive activity, physical activity, and social activity)) [50]. In addition, those at risk of social isolation who lived with others were reported to be significantly more likely to be male, to have only a public pension as income, to have fewer IADL impairments, and to have less fear of falling compared with those who lived alone and were at risk for social isolation (social isolation assessed using the Lubben Social Network Scale (LSNS-6)) [33]. The previous studies described above suggest the possibility that the factors associated with loneliness and social isolation differ with household type, such as whether an individual lives alone or with their family. However, we found no studies that measured and compared loneliness and social isolation using reliable and valid scales, identically with household type.

Therefore, the purpose of this study was to clarify the actual conditions of loneliness and social isolation simultaneously using household type, whether an individual lives alone (single-person type: ST) or with their family (multi-person type: MT), and to identify the factors related including self-efficacy to loneliness and social isolation by household type.

## 2. Materials and Methods

### 2.1. Study Design

This study used a cross-sectional design with an anonymous, self-administered survey.

### 2.2. Study Participants and Setting

The target population was independent, community-dwelling older adults. We aimed to survey a representative sample of up to 5351 older adults (aged 65–75 years), based on the 5351 community general support centers throughout Japan. To achieve this, all centers were contacted and asked to provide the questionnaire to one randomly selected community member who met the following inclusion criteria: (1) aged ≥ 65 years; (2) living in the community (i.e., not in a hospital or residential care facility); and (3) living as an independent older adult (i.e., no requirement for long-term care or support). These criteria were in accordance with the Certified Level of Need for Long-Term Care National Insurance of Japan (Kaigo Hoken in Japanese).

Data were collected between July and August 2022 using anonymous self-administered paper questionnaires. Completed questionnaires were returned by mail to the academic office at Hokkaido University by the older adults. A total of 1336 (25.0%) participants completed the questionnaire, and 1115 (20.8%) were included in the final analysis after the exclusion of those aged <65 years, those with missing data regarding living status, and those who answered “other” regarding living status.

### 2.3. Measures

#### 2.3.1. Loneliness

The three-item Japanese version of the University of California Los Angeles Loneliness Scale version 3 (UCLA-LS3-J SF-3 [51]) was used to evaluate loneliness. In this scale, which is based on the theory of Peplau et al. [5], loneliness is defined as “the unpleasant experience that occurs when a person’s network of social relationships is deficient in some important way, either quantitatively or qualitatively” [11]. The UCLA scale includes the following three items: (1) “How often do you feel that you lack companionship?”, (2) “How often do you feel left out?”, (3) “How often do you feel isolated from others?”. Each of the three items in the UCLA-LS3-J SF-3 [51,52] has four response choices: (1) never, (2) rarely, (3) sometimes, and (4) always. Total scores range from 3 to 12, with higher scores indicating a higher level of loneliness. The Cronbach’s alpha of the Japanese-translated version of the scale was 0.790 [51].

#### 2.3.2. Social Isolation

The Japanese version of the Lubben Social Network Scale (LSNS-6) was used to evaluate social isolation [15,37]. In this scale, which is based on the theory of Lubben, social isolation is defined as a “lack of redundancy of social ties” [15]. The LSNS-6 includes the following six items: (1) “How many relatives do you see or hear from at least once a month?”, (2) “How many relatives do you feel close to such that you could call on them for help?”, (3) “How many relatives do you feel at ease with that you can talk about private matters?”, (4) “How many of your friends do you see or hear from at least once a month?”, (5) “How many friends do you feel close to such that you could call on them for help?”, (6) “How many friends do you feel at ease with that you can talk about private matters?”. Each of the six items in the LSNS-6 has six choices: “none” (0 points), “one” (1 point), “two” (2 points), “three or four” (3 points), “five to eight” (4 points), and “nine or more” (5 points). Total scores range from 0 to 30, with higher scores indicating a larger social network. The Cronbach’s alpha of the Japanese-translated version of the scale was 0.82 [37].

#### 2.3.3. Self-Efficacy

The General Self-Efficacy Scale (GSES), which has been validated for reliability and validity, was used to evaluate perceived self-efficacy as high or low [53]. Each of the 16 items in the GSES was answered with a two-item choice response (yes/no). Total scores ranged from 0 to 16, with higher scores indicating greater self-efficacy.

#### 2.3.4. Demographic Variables

The demographic characteristics of interest included age, gender, living status, educational level, subjective economic status, disease treatment status, and a single question about subjective health status. Gender was classified as male (1) or female (2). Living status was classified as living alone (1), living with a spouse (2), living with spouse and children (3), living with children and grandchildren (4), or other (5). The educational level was classified as elementary school/junior high school (1), high school (2), vocational school (3), university (4), graduate school (5), or other (6). Subjective economic status was rated on a four-point, Likert-type scale: 1 = I am not in trouble, 2 = I am not in much trouble, 3 = I am in a little trouble, and 4 = I am in trouble. Disease treatment status was classified as yes (0) or no (1). Subjective health status was rated on a four-point, Likert-type scale: 1 = Very healthy, 2 = Quite healthy, 3 = Not very healthy, and 4 = Not at all healthy. The subjective health status scores were reversed using IBM SPSS Statistics (ver. 26.0; IBM Corp., Armonk, NY, USA), such that higher scores indicate better health.

### 2.4. Statistical Analysis

On the basis of living status, participants were classified into two groups: single-person type (ST) and multi-person type (MT). ST included individuals who responded as living alone (1) with respect to living status, and MT included individuals who responded as living with a spouse (2), living with spouse and children (3), and living with children and grandchildren (4) with respect to living status. First, we used chi-square tests and t-tests to assess the differences between ST and MT. Second, to assess characteristics of loneliness and social isolation using the household type of community-dwelling older adults, we carried out covariance analysis with age and gender as covariate variables. Third, to examine the factors that influenced loneliness and social isolation, we carried out multiple regression analysis (forced entry method). *p* < 0.05 was considered to indicate statistical significance, and IBM SPSS Statistics (ver. 26.0) was used for all statistical analyses.

### 2.5. Ethical Considerations

This research was conducted in accordance with the 1964 Declaration of Helsinki (and its amendments), and the ethical guidelines for life sciences and medical research involving human subjects presented by the Ministry of Health, Labour and Welfare of Japan. The study protocol was approved by the Institutional Review Board of Hokkaido University (approval No. 21–91, 31 March 2022). All subjects provided written informed consent and voluntarily participated in this study after receiving a written explanation of its purpose and methods, as well as of ethical and all other aspects relevant to their decision to participate.

## 3. Results

### 3.1. Demographic Characteristics

Table 1 shows the demographic characteristics of the participants. The mean age was 74.92 years (standard deviation (SD) = 6.68) for ST participants and 71.76 years (SD = 4.90) for MT participants. In total, 63.5% of ST and 51.5% of MT participants were female, and 44.3% of ST and 49.5% of MT participants had a high school level of education. Additionally, 73.3% of ST and 84.0% of MT participants indicated that they were not “in trouble” with respect to their subjective economic status, 89.2% of ST and 79.2% of MT participants had a medical condition for which they were currently being treated, and 71.0% of ST and 86.8% of MT participants indicated that their subjective health status was “healthy”. The mean GSES was 8.56 (SD = 4.30) for ST and 9.82 (SD = 4.07) for MT participants. The *t*-test and chi-square test showed significant differences in all items for both groups (ST and MT).

### 3.2. Characteristics of Loneliness and Social Isolation by Household Type

Table 2 shows the characteristics of loneliness by household type. The mean UCLA-LS3-J SF-3 scores were 6.32 (SD = 2.49) for ST and 5.62 (SD = 1.95) for MT. There was a significant UCLA-LS3-J SF-3 point difference by household type (*p* < 0.001). After adjusting for age, gender, and age × gender, there was a significant difference in UCLA-LS3-J SF-3 scores between ST and MT, with higher scores in ST.

Table 3 shows the characteristics of social isolation by household type. The mean LSNS-6 scores were ST = 12.97 (SD = 7.01) and MT = 17.02 (SD = 5.97). There was a significant difference in LSNS-6 scores by household type (*p* < 0.001). After adjusting for age, gender, and age × gender, there was a significant difference in LSNS-6 scores between ST and MT, with lower scores in ST.

### 3.3. Factors Associated with Loneliness and Social Isolation by Household Type

The factors that influenced loneliness are shown in Table 4. Receiving treatment for a disease/condition was not included in the multiple regression analysis because simple linear regression analysis for both ST and MT showed no significant association between UCLA score and receiving treatment for a disease/condition (ST, *p* = 0.809; MT, *p* = 0.155). In ST, high loneliness was associated with educational level (β = 0.104, *p* = 0.045), poor subjective economic status (β = 0.165, *p* = 0.002), low subjective health status (β = −0.140, *p* = 0.010), and low GSES scores (β = −0.476, *p* < 0.001). The adjusted R^2^ was 0.366 (Table 4). In MT, high loneliness was associated with poor subjective economic status (β = 0.124, *p* < 0.001), low subjective health status (β = −0.105, *p* = 0.003), and low GSES scores (β = −0.381, *p* < 0.001). The adjusted R^2^ was 0.222 (Table 4). The GSES had a greater impact on loneliness in both ST and MT.

Table 5 shows the factors related to social isolation. Receiving treatment for a disease/condition was not included in the multiple regression analysis because simple linear regression analysis for both ST and MT showed no significant association between the LSNS-6 score and receiving treatment for a disease/condition (ST, *p* = 0.078; MT, *p* = 0.529). In ST, a larger social network was associated with gender (β = 0.202, *p* < 0.001), high subjective health status (β = 0.202, *p* < 0.001), and high GSES scores (β = 0.358, *p* < 0.001). The adjusted R^2^ was 0.310 (Table 5). In MT, a larger social network was associated with gender (β = 0.073, *p* = 0.035), good subjective economic status (β = −0.087, *p* = 0.016), high subjective health status (β = 0.105, *p* = 0.004), and high GSES scores (β = 0.295, *p* < 0.001). The adjusted R^2^ was 0.145 (Table 5). The GSES had a greater impact on social isolation in both the ST and MT groups.

## 4. Discussion

In this study, we clarified the factors related to loneliness and social isolation in older adults using household type (ST or MT) among 5351 Japanese older adults (aged 65 years or older) using a national survey.

### 4.1. Comparison between UCLA and LSNS-6 by Household Type

In this study, ST and MT participants had mean UCLA scores of 6.32 and 5.62, with ST participants scoring significantly higher than MT participants, even when gender and age were taken into account. Previous studies have identified living alone as a factor related to loneliness among older adults [54], and in the present study, ST participants’ UCLA scores were significantly higher. Therefore, the results indicate the importance of focusing on household type when assessing loneliness among older adults. In addition, in this study, the mean LSNS-6 scores were 12.97 for ST and 17.02 for MT participants, and MT participants’ scores were significantly higher than those of ST participants, even when gender and age were taken into account. Living alone has been found to be a predictor for social isolation [46], and, in this study, LSNS-6 scores were significantly lower for ST participants. Thus, the results indicate the importance of focusing on household type when assessing social isolation. No previous study has conducted comparisons between loneliness and social isolation among older adults measured using household type with a reliable and valid scale. Therefore, the findings from this study provide important comparison data.

### 4.2. Comparison of UCLA-Related Factors by Household Type

The results of the multiple regression analysis showed that ST had significant associations with UCLA for educational level, subjective economic status, subjective health status, and GSES, while MT had significant associations with UCLA for subjective economic status, subjective health status, and GSES. Studies conducted before the coronavirus pandemic revealed that health problems (multiple diseases, poor health) [55,56], subjective health status [35], low income [35,55], and education (years) [34] were associated with loneliness, which is consistent with the results of the present study. Therefore, variables such as educational level, subjective economic status, and subjective health status may be important variables for predicting loneliness in ST and MT individuals during the coronavirus pandemic. Regarding age and gender, there was no significant association with UCLA scores for ST and MT participants. Regarding the relationship between age and loneliness, being 75 or older has been identified as a risk factor for loneliness [32]. However, it has also been shown that loneliness levels off after the age of 90 [57], so the results regarding the relationship between age and loneliness are not consistent. In addition, a meta-analysis examining gender differences in lifetime loneliness found similar levels of loneliness among men and women throughout life, supporting the results of this study [58]. The current results indicate the importance of variables other than age and gender as factors associated with loneliness in ST and MT participants.

### 4.3. Comparison of LSNS-6 Related Factors by Household Type

The results of the multiple regression analysis showed that, for ST participants, significant associations with LSNS-6 were found for gender, subjective health status, and GSES. For MT participants, significant associations were found for gender, subjective economic status, subjective health status, and GSES. In addition, for both ST and MT participants, social isolation had the strongest association with GSES. Studies conducted before the coronavirus pandemic showed that having a low monthly income [31,36] and being male [31] were risk factors for social isolation. In addition, a trend toward higher mean LSNS-6 scores was observed in a group of participants with good subjective health [37], which is consistent with the results of this study. Therefore, variables such as gender, economic status, and subjective health status may be important for predicting social isolation for ST and MT individuals during the coronavirus pandemic. It is possible that women may have larger social networks for both ST and MT, although the association between gender and social isolation was greater for ST (ST (β = 0.202, *p* < 0.001); MT (β = 0.073, *p* = 0.035)). A survey among older adults in Japan found that just 6.6% of older women living alone rarely socialized with their neighbors compared with 17.4% of older men living alone [59]. Older men who live alone, therefore, have fewer interactions with other people in the local community. This may be because of differences in gender roles between men and women. When asked in a study whether they agreed or disagreed with the idea that “husbands should work outside the home and wives should take care of the home”, 33.6% of those aged 60–69 years old and 46.1% of those aged 70 years and older agreed [60]. These results show that gender roles are often viewed in a fixed way among older adults in Japan. Women may have larger social networks than men as a result of their activities in a wide range of areas, including family and community. In addressing social isolation among older adults, it is therefore important to consider both gender and household type. In contrast, in the present study, subjective economic status, which was significantly associated with social isolation for MT participants, was not significantly associated with social isolation for ST participants, which is different from the results of previous studies [31,36]. However, because some studies have reported that low income is not associated with the risk of social isolation [33], the association between economic status and social isolation is not consistent. Regarding age, there was no significant association with LSNS-6 scores for ST and MT. One study reported that the risk of social isolation increases with age [33], which is a finding that differs from some previous studies. The results of this study indicate the importance of variables other than age as factors related to social isolation for ST and MT individuals.

### 4.4. Future Research

Our findings on the association between GSES and loneliness and social isolation in ST and MT are consistent with previous reports [45,61] indicating that low self-efficacy is a predictor for loneliness and social isolation. However, previous studies that have identified the relationship between self-efficacy and loneliness and social isolation have focused on Chinese adults aged 18 and older [41], Polish adults aged 18 and older [42], Dutch adults aged 55 and older [43], Taiwanese adults aged 65 and older [44], and Chinese adults aged 60 and older [45]. There have, therefore, been few studies on older adults aged 65 and older. This study is valuable because it shows that increasing self-efficacy is important for both loneliness and social isolation among older adults aged 65 and older. According to a conceptual analysis of self-efficacy [62], the first consequence of gaining self-efficacy is the achievement of behavior. Someone with a higher self-efficacy for a task is more likely to accomplish it [63]. The second is an effort toward achievement. Higher self-efficacy is associated with a greater tendency to strive to challenge the target behavior [64]. Third, there are changes in physiological and psychological responses. Anxiety and fear tend to be weaker when self-efficacy is high [65]. Self-efficacy is therefore an excellent predictor for behavior and behavioral change. Low self-efficacy may result in underachievement of behavior, inadequate effort to behave in particular ways, and changes in physiological and psychological reactions (with stronger manifestations of anxiety and fear), which may lead to loneliness and social isolation. Our findings suggest that there is a need to develop healthcare systems and programs based on self-efficacy using household type to address both loneliness and social isolation. The antecedent requirements of self-efficacy include meaningfulness and the need for action. When people place more value on the meaning of what they do, they have higher self-efficacy and rates of achieving the desired behavior [66,67]. Other factors that have been found to increase self-efficacy include knowing and being able to use strategies to accomplish certain tasks [68]. To prevent loneliness and social isolation, it is, therefore, necessary to focus on these prior requirements of self-efficacy and expand community care and self-care. In addition, GSES is a scale that measures the strength of general self-efficacy shown by individuals in their daily lives [53]. It is difficult to evaluate task-specific self-efficacy. In the future, it will therefore be necessary to clarify the concept of self-efficacy, which has a specific impact on issues such as loneliness and social isolation, and to develop a scale to measure this concept.

### 4.5. Limitations

The current study involved several limitations. First, because this was a cross-sectional study, the causal relationships between loneliness and social isolation and related factors such as GSES are unclear. Future longitudinal studies should be conducted to track UCLA and LSNS-6 scores in ST and MT individuals over time. Second, although focusing on GSES as a factor related to UCLA and LSNS-6 is a strength of the current study, the adjusted R^2^ value was low, and other factors may not have been adequately taken into account. Third, the background of ST participants was not taken into account. Previous studies have reported that loneliness is significantly associated with being divorced [69]. The specific backgrounds of ST individuals, such as divorce, bereavement, and being unmarried, should be considered in future studies.

## Figures and Tables

**Table 1 healthcare-11-01647-t001:** Demographic characteristics of the participants.

	ST N = 296	MT N = 819	*p*
Variables	mean ± SD		
Age (years)	74.92 ± 6.68	71.76 ± 4.90	<0.001
GSES	8.56 ± 4.30	9.82 ± 4.07	<0.001
Variables	N (%)		
Gender			<0.001
Male	106 (35.8)	396 (48.4)	
Female	188 (63.5)	422 (51.5)	
Missing data	2 (0.7)	1 (0.1)	
Educational level			<0.001
Elementary school/junior high school	65 (22.0)	72 (8.8)	
High school	131 (44.3)	405 (49.5)	
Vocational school	52 (17.6)	153 (18.7)	
University	42 (14.2)	168 (20.5)	
Graduate school	1 (0.3)	11 (1.3)	
Other	1 (0.3)	10 (1.2)	
Missing data	4 (1.3)	0	
Subjective economic status			<0.001
I am not in trouble	109 (36.8)	337 (41.1)	
I am not in much trouble	108 (36.5)	351 (42.9)	
I am in a little trouble	60 (20.3)	116 (14.2)	
I am in trouble	19 (6.4)	14 (1.7)	
Missing data	0	1 (0.1)	
Receiving treatment for a disease/condition			<0.001
Yes	264 (89.2)	649 (79.2)	
Subjective health status			<0.001
Very healthy	23 (7.8)	99 (12.1)	
Quite healthy	187 (63.2)	612 (74.7)	
Not very healthy	61 (20.6)	83 (10.1)	
Not at all healthy	22 (7.4)	21 (2.6)	
Missing data	3 (1.0)	4 (0.5)	

SD, standard deviation. Chi-square tests and *t*-tests.

**Table 2 healthcare-11-01647-t002:** Characteristics of loneliness by household type for community-dwelling older adults (N = 1115).

	ST	MT	F-Value	*p*
Mean ± SD ^a^	6.32 ± 2.49	5.62 ± 1.95	―	<0.001
Age-adjusted mean ± SD ^b^	6.31 ± 2.49	5.61 ± 1.94	23.19	<0.001
Gender-adjusted mean ± SD ^b^	6.32 ± 2.49	5.62 ± 1.95	27.06	<0.001
Age- × gender-adjusted mean ± SD ^b^	6.31 ± 2.49	5.61 ± 1.94	26.36	<0.001

^a^: *t*-test. ^b^: analysis of covariance.

**Table 3 healthcare-11-01647-t003:** Characteristics of social isolation by household type for community-dwelling older adults (N = 1115).

	ST	MT	F-Value	*p*
Mean ± SD ^a^	12.97 ± 7.01	17.02 ± 5.97	―	<0.001
Age-adjusted mean ± SD ^b^	13.04 ± 7.00	17.04 ± 5.97	80.80	<0.001
Gender-adjusted mean ± SD ^b^	12.99 ± 7.01	17.02 ± 5.97	98.76	<0.001
Age- × gender-adjusted mean ± SD ^b^	13.06 ± 7.01	17.03 ± 5.97	90.32	<0.001

^a^: *t*-test. ^b^: analysis of covariance.

**Table 4 healthcare-11-01647-t004:** Factors associated with loneliness by household type for community-dwelling older adults (multiple regression analysis: forced entry method).

	ST N = 266		MT N = 764	
	β	*p*	β	*p*
Age	−0.043	0.391	0.016	0.635
Gender	−0.086	0.087	−0.010	0.756
Educational level	0.104	0.045	0.043	0.193
Subjective economic status	0.165	0.002	0.124	<0.001
Subjective health status	−0.140	0.010	−0.105	0.003
GSES	−0.476	<0.001	−0.381	<0.001
R^2^	0.380		0.228	
Adjusted R^2^	0.366		0.222	

**Table 5 healthcare-11-01647-t005:** Factors associated with social isolation by household type for community-dwelling older adults (multiple regression analysis: forced entry method).

	ST N = 265		MT N = 754	
	β	*p*	β	*p*
Age	0.002	0.972	−0.004	0.906
Gender	0.202	<0.001	0.073	0.035
Educational level	−0.017	0.748	−0.017	0.619
Subjective economic status	−0.099	0.078	−0.087	0.016
Subjective health status	0.202	<0.001	0.105	0.004
GSES	0.358	<0.001	0.295	<0.001
R^2^	0.325		0.152	
Adjusted R^2^	0.310		0.145	

## Data Availability

The datasets used or analyzed during the current study are available from the corresponding author upon reasonable request.

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
