# Peer review of "Differences in Loneliness and Social Isolation among Community-Dwelling Older Adults by Household Type: A Nationwide Survey in Japan"

_healthcare, 2023, doi:10.3390/healthcare11111647_

Round 1

Reviewer 1 Report

Introduction:
One variable that can be used as a measure of cognition is self-efficacy. The authors should explain in more detail how self-efficacy can be used to measure cognition in the context of loneliness and social isolation. In particular, they should discuss possible causal relationships and explain the assumptions underlying the study (lines 79-85).
Methods:
Were other health-related data collected? If so, why was this not considered as possible influencing factors.
Results:
Taking subjective health status into account when assessing loneliness and social isolation (table 3 and table 4) would be an interesting addition. Why was no combined model calculated for loneliness and social isolation?
Discussion:
The relationship between loneliness and social isolation and self-efficacy should be discussed in more detail. Furthermore, further explanations for the age and especially gender differences found should be integrated and related to the context of Japanese society.

ok

Reviewer 2 Report

interesting issue. 

Study found that , loneliness  is more, social network is lesser, and self efficacy is low in those living alone. Effect of education, economic status, health status are also considered.

Comment 01: Loneliness can be subjective. and there is not accepted definition. 

Reviewer 3 Report

This study tested whether loneliness and social isolation was sign different between older adults living in a single-person type (ST) vs. in a multi-person type (MT). It is impressive to see that the authors collected data from a self-administered survey with 5,351 Japanese.The results demonstrate differences in both DVs, which is nice. However, I am missing a theoretical backdrop of the study which could also guide the authors with testing for interactions between gender and other factors or testing non-linear relationships. I would strongly recommend reviewing theory describing / reviewing papers such as 

Lippke, S. & Warner, L. (2023). Understanding and overcoming challenges in times of personal or global crisis – Editorial on the Special Issue on Loneliness and Health. Applied Psychology: Health and Well-Being, 15(1), 3-23. https://doi.org/10.1111/aphw.12420 (Open Access: https://iaap-journals.onlinelibrary.wiley.com/doi/epdf/10.1111/aphw.12420)

and then testing for more aspects. Accordingly, all sections need to be revised, including the discussion. Please also make sure that your discussion is more concrete and not just on a superficial level, including replacing the sentence "New strategies should be developed for applying GSES to reduce social isolation and loneliness according to household type."

However, I have one more concern the authors have to address, too: If this paper should be considered in the journal Healthcare, the regarding link needs to be explained more explicitly. Currently, I like the paper in general, but it does not fit at all with the current journal.

ok
